# Effects of Fermented Manure Bedding Thickness on Bulls’ Growth, Behavior, and Welfare as Well as Barn Gases Concentration in the Barn

**DOI:** 10.3390/ani12070925

**Published:** 2022-04-04

**Authors:** Kaifeng Niu, Xinxin Zhang, Chao Chen, Liguo Yang

**Affiliations:** 1Key Lab of Agricultural Animal Genetics, Breeding and Reproduction of Ministry of Education, College of Animal Science and Technology, Huazhong Agricultural University, Wuhan 430070, China; nkf_19930806@163.com (K.N.); zxx1144795936@163.com (X.Z.); chenchao1995@webmail.hzau.edu.cn (C.C.); 2Hubei Province’s Engineering Research Center in Buffalo Breeding and Products, Wuhan 430070, China

**Keywords:** fermented manure, average daily gain, behavior, animal welfare, bedding

## Abstract

**Simple Summary:**

Due to the increasing cost of common bedding materials, there is a growing preference for recycled manure solids as bedding materials. This study aimed to evaluate bull growth performance, behavior, and animal welfare, and gases concentration in the barn under different thicknesses of bedding. Our results indicated that deep fermented bedding promoted the growth and welfare of bulls. The results showed that soft fermented bedding played a positive role in the growth and development of bulls, and it increased the feed conversion rate (F/G, ratio of feed to gain), and the effect of deep fermented bedding was better than the shallow one. The results indicated that the DFB (deep fermented bedding) group exhibited optimal hoof health, body hygiene, and lying time, followed by the SFB (shallow fermented bedding) group and the CF (concrete floor) group. As for the gases concentration, the contents of ammonia and carbon dioxide were the lowest in the DFB group, followed by the SFB group, and they were the highest in the CF group at the same time points. In summary, fermented manure bedding significantly improves the growth performances, behavior, and welfare of bulls as well as gases concentration in the barn, and the improvement effect achieved by deep fermented bedding is more obvious than by shallow fermented bedding.

**Abstract:**

Providing clean, comfortable bedding is essential for the growth and welfare of bulls. This study was aimed to investigate the effects of bedding thickness on growth performance, behavior, and welfare of bulls as well as gases concentration in the barn. Thirty-six healthy Simmental bulls (7–9 months old) were randomly divided into three groups and raised on 0 cm (concrete floor, CF), 15 cm (shallow fermented bedding, SFB), and 30 cm (deep fermented bedding, DFB) fermented manure bedding. The results showed that the DFB group exhibited the optimal ADG (average daily gain), F/G (ratio of feed to gain), hoof health, body hygiene, and lying time, followed by the SFB group and the CF group (*p <* 0.05). As for the barn gas environment, the contents of ammonia and carbon dioxide were the lowest in the DFB group, followed by the SFB group, and they were the highest in the CF group at the same time points (*p <* 0.01). In summary, fermented manure bedding significantly improves the growth performances, behavior, and welfare of bulls as well as gases concentration, and the improvement effect achieved by deep fermented bedding is more obvious than by shallow fermented bedding.

## 1. Introduction

The most commonly used bedding materials in barn systems are sawdust and sand [1]. Other materials such as straw, peanut shells, and wood chips are also commonly used as bedding materials [2,3,4,5]. The increasing demand for bedding materials, especially sawdust and wheat straw, leads to the rising cost of conventional bedding materials [6]. Due to the increasing cost of common bedding materials, there is a growing preference for recycled manure solids as bedding materials [7]. The recycled manure solids tend to be used as bedding materials to form deep bedding and shallow bedding [8].

Research shows that bedding plays a crucial role in improving bovine comfort and bovine lying behavior [9]. Cows lie longer when bedding materials are soft and dry [10]. The daily weight gain of bulls raised on soft rubber mats increased by 9.09% in comparison with that of bulls raised on a concrete floor [11]. Cows prefer lying down on a concrete floor covered with a large amount of straw to a floor covered with a soft rubber bed, and they spend more time lying in deep-bedded sawdust (15 cm or above) and deep-bedded sand stalls than in shallow-bedded sawdust (2–3 cm) stalls [12,13,14]. In addition, the increase in sawdust bedding material on geotextile mattresses positively affects the lying preferences of cows [15]. Each additional kilogram of straw results in an increase in cows’ daily lying time by 12 min. Correspondingly, with every 1 cm decrease in bedding thickness, cows spend 11 min less lying down during each 24 h period [16]. One previous study has shown that farms using compost-bedded pack systems (CBS) exhibited 13.3% higher milk production per cow than farms using a dry lot system (DLS) [17]. Another study reported that the application of a compost-bedded pack system significantly improves the welfare and comfort of cows [18].

Compared with the bulls raised on concrete floors, the bulls raised on rubber mats lie longer and display higher cleanliness and lower joint swelling risk [12,19]. The deep bedding sheds with wood chips or straw are preferred by cows, which contributes to the decline in hock joint disease [20]. Similarly, cows raised on deep sand bedding have been reported to have less hock damage than cows raised on mattresses [21,22]. Deep bedding such as sand bedding may provide better lying comfort for lame cows than an unbedded rubber surface [23]. The use of deep-bedded manure solids in a free-stall barn reduces the incidence of lameness and hock abrasions [7]. Barrientos et al. [24] surveyed 76 farms and found that bedding depth of at least 10 cm reduced the risk of hock injury. Deep sand or sawdust bedding provides limping cows with a more comfortable environment than conventional rubber bedding, thus prolonging lying time [25,26,27]. Concrete floor significantly increased the severity of joint injury in cattle, compared with deep sand bedding or deep compost bedding [28]. 

The aim of this study was to compare the effects of bedding thickness on bull growth performance, behavior, welfare, and gases concentration in the barn. We hypothesized that fermented manure bedding would positively influence the growth, hock health, hygiene, and lying time of bulls, and gases concentration in the barn.

## 2. Materials and Methods

The study was conducted at the Hanjiang Cattle Industry Co., Ltd., Jingmen China, from July 2019 to January 2020. The protocol of this experiment was approved by the Scientific Ethic Committee of Huazhong Agricultural University (HZAUCA-2017–011) and the animal trial was conducted in accordance with the National Institute of Health Guidelines for the Care and Use of Experiment Animals (Beijing, China).

### 2.1. Manufacture of Fermented Manure Bedding

The manufacturing process of cow manure into bedding material through a harmless fermentation treatment is shown in Figure 1. Firstly, fresh bovine manure and rice chaff were collected and mixed by a loader, and the moisture content of the mixture was adjusted to about 60%. Then, the microbial agent was evenly sprinkled at 1 kg/m^3^ and mixed to build a strip heap with a length of 15 m, a width of 7 m, and a height of 1.5 m. The microbial agent mainly contained bacillus subtilis DK068, lactic acid bacteria, yeast MX0126, cellulase, and lignin enzyme, with viable bacteria count >1.0 × 10^10^ CFU/g.

The temperature of the pile was detected with a handheld digital display thermometer every day. When the temperature of the pile reached 55 °C, the pile was stirred with a forklift every 5 days. The bovine manure underwent the heating period, high-temperature period, cooling period, and finally, the pile temperature fell to room temperature, indicating the completion of harmless ectopic fermentation. The fermented bovine manure was bedded to the cattle pen with a forklift. the CF group was bedded with no bovine manure. The SFB group was bedded with a thickness of 15 cm fermented bovine manure. The DFB group was bedded with a thickness of 30 cm fermented bovine manure.

### 2.2. Animal Management

A total of 36 healthy Simmental bulls (body weight: 271.40 ± 6.55 kg; age: 7–9 months) were randomly divided into three groups (12 cattle per group). The cowshed was a semi-enclosed barn with pens, and the water tank was located outside the barn. Three independent pens within the same barn were respectively assigned to three bedding treatment groups including concrete floor (CF group), shallow fermented bedding (15 cm in thickness, SFB group), and deep fermented bedding (30 cm in thickness, DFB group). The bedding material was fermented bovine manure. The area of each pen was not less than 120 m^2^, and the movement area of each head was not less than 10 m^2^. During 50 days of the experimental period, the animals were fed with a total mixed ration twice daily (at 09:00 and 16:00) with free access to feed and water. All bulls were fed the same diet during the experiment, and the specific nutrients were shown in Table 1. The CF group pen was cleaned daily, and the SFB and DFB group pens were regularly supplemented with fermented bedding.

### 2.3. Productive Performance

Average daily gain (ADG): At the beginning and end of the experiment, the bulls were weighed at an empty stomach state, and the average daily gain was calculated by subtracting the weight at the end of the experiment from the weight at the beginning of the experiment and dividing by the number of days of the experiment.
Average daily gain (ADG) = (end weight − initial weight)/days

Average daily feed intake (ADFI): The total amount of feed and residual amount of each group were recorded for 3 consecutive days a week during the trial, and the average daily feed intake of each cow was calculated in this way: Average daily feed intake (ADFI) = (total feed weight − residual feed weight)/12

Feed to gain ratio (F/G): Each treatment group’s feed-to-gain ratio is the ratio of average daily feed intake to average daily gain of each treatment group.
F/G = ADFI/ADG

### 2.4. Behavioral Index

A total of 36 bulls in three cattle pens were continuously observed for 8 h on a daily basis (8:00–16:00) by camera and human observation. The digital cameras were installed opposite to each pen (with one camera corresponding to one pen). The behavior observation was conducted once a week for 7 consecutive weeks. The observation mainly focused on the following items: 

Lying: the belly of the body is in contact with the ground, and the body is supported by the ground rather than the hooves and legs.

Preparation time before lying down: the time it takes to sniff the ground for position selection and lie down. 

Standing: the body is supported by at least three legs on the ground.

Feeding: the feed is sucked into the mouth with tongue.

Rumination: feed is inversely vomited to re-enter the oral cavity and chewed again before swallowing.

### 2.5. Animal Welfare Index

During the experiment, the three groups’ hygiene score, locomotion score, and hock lesion score of the bulls were scored individually every 10 days by 1 trained observer, 6 times in total.

The hygiene score was assessed by the amount of dirt on the udder and lower hind legs based on a 4-point scale with 1 = clean and 4 = dirty [29]. The average body condition and hygiene score were calculated for each pen for analysis. 

Bulls were evaluated for lameness using a 5-point locomotion scoring method [30]. Locomotion score (LS) was as follows: 1 = normal locomotion, 2 = imperfect locomotion, 3 = lame, 4 = moderately lame, and 5 = severely lame. 

The severity of back leg hock lesion (HL) was measured using the 6-point scale scoring system as described previously [31]. Hock lesions were classified as 1 = no lesion, 3 = hair loss (mild lesion), and 6 = swollen hock with hair loss (severe lesion). 

### 2.6. Gases Concentration in the Barn

The concentrations of carbon dioxide and ammonia in the cowshed were measured by a portable carbon dioxide tester and a portable ammonia detector at a distance of 0.7 m above the bedding at 7:00, 12:00, and 18:00, every day. A set of “Z”-shaped equidistant 5 measurement points were selected. The basic information of the detectors is shown in Table 2. The surface temperature of the bedding was measured at the same place and time every day with a digital display thermometer.

### 2.7. Statistical Analysis

Statistical analysis was performed by SPSS software (SPSS v. 21, SPSS Inc.; Chicago, IL, USA). Significance analysis based on the production performance, behavioral index, animal welfare indicators, and cowshed gas environment index were conducted by one-way ANOVA in SPSS. *p* < 0.05 was used to indicate a significant difference.

## 3. Result

### 3.1. Productive Performance

As presented in Table 3, compared with the CF group, the average daily gain (ADG) of bulls in the DFB group and the SFB group increased by 13.63% and 7.21%, but there was no significant difference in daily gain between the three groups (*p* > 0.05). There was no difference in average daily feed intake (ADFI) among the three groups (*p* > 0.05), but the daily feed intake of the CF group was higher than the other two groups (Table 3). It is shown that the ratio of feed to gain in the DFB group was 20.93% lower than that in the CF group, and the ratio of feed to gain in the SFB group was 16.76% lower than that in the CF group; the difference was significant (*p* < 0.05). There was no significant difference in F/G between the SFB group and the DFB group (*p* > 0.05) (Table 3).

### 3.2. Behavioral Index

The influence of bedding thickness on bull behavior is shown in Table 4. As presented in Table 4, the indicators of lying time, rumination time, and rumination frequency of bulls in the fermented bedding group were significantly higher than those in the CF group (*p* < 0.01), and the DFB group was significantly higher than the SFB group (*p* < 0.01). The frequency of lying and standing in the fermented bedding group was significantly higher than that in the CF group (*p* < 0.01), but there was no significant difference between the DFB group and the SFB group (*p* > 0.05). The standing time of bulls in CF group was significantly higher than that in the fermented bedding group (*p* < 0.01), and the standing time of bulls in the SFB group was significantly higher than that in the DFB group (*p* < 0.01). In terms of intake time, intake frequency, and preparation time before lying, the CF group was significantly higher than the fermented bedding group (*p* < 0.01), but there was no difference between the DFB group and the SFB group (*p* > 0.05) (Table 4).

### 3.3. Animal Welfare Index

As can be seen from Figure 2, the body surface hygiene of bulls in the CF group was the worst, and the score was significantly higher than the other two groups (SFB and DFB) (*p* < 0.01). The body surface condition of bulls in the DFB group was the best, and the score was significantly lower than that in the SFB group (*p* < 0.01). The hock and locomotion scores were measured based on a 6-point scale (0–5) and 5-point scale (1–5), respectively. A higher score means a worse health condition. The severity of hock lesions injury in the CF cattle was significantly (*p* < 0.01) higher than the DSB and the SFB cattle, the locomotion score of the CF cattle was the highest among the three groups (Figure 2). 

In terms of locomotion score and hock lesion, at the end of the experiment, one cow in the CF group had a severe limp (5 points), three cattle had moderate limps (3 points), and two cattle had mild limps (2 points). In contrast, all cattle in the SFB and DFB groups had a locomotion score of 1 point without limp until the end of the experiment. None of the cattle had severe hock lesions, and 5 cattle in the CF group had minor hock lesions. Two cattle in the SFB group had small hock lesions. Only 1 cow in the DFB group had a minor hock lesion.

### 3.4. Gases Concentration in the Barn

It can be seen from Table 5 that the carbon dioxide content in the SFB and DFB groups at 12:00 is significantly lower than in the CF group (*p* < 0.01) (Table 5). There was no difference in carbon dioxide content between the SFB group and the DFB group (*p* > 0.05). There was no difference in carbon dioxide content between the three groups at 7:00 and 18:00 (*p* > 0.05), but the carbon dioxide content in the CF group was the highest, while that in the DFB group was the lowest. As can be seen from Table 5, at the same time point (7:00, 12:00, and 18:00), the ammonia content in the SFB group and the DFB group was significantly lower than that in the CF group. The SFB group and DFB group were significantly lower than that in the CF group (*p* < 0.01); the 18:00 SFB group and DFB group were significantly lower than the CF group (*p* < 0.05). At the same time point (7:00, 12:00, and 18:00), there was no difference in ammonia content between the SFB group and the DFB group (*p* > 0.05). Still, the ammonia content in the DFB group was slightly lower than that in the SFB group (Table 5). At the same time point (7:00, 12:00, and 18:00), the surface temperature in the CF group was significantly lower than the 2 fermented bedding groups, and the surface temperature of the bedding in the DFB group was significantly higher than the SFB group (Table 5).

## 4. Discussion

This study aimed to evaluate bull growth performance, behavior, and animal welfare, and gases concentration under different thicknesses of bedding. Our results indicated that deep fermented bedding promoted the growth and welfare of bulls.

The use of comfortable soft bedding with adequate materials in the barn provided a quality environment for cattle’s rest and growth [7]. Holstein heifers (225–400 kg) raised on soft rubber mats increased their average daily gain by 9.09% compared with those raised on concrete floors [11]. The average daily gain and expected carcass average daily gain of 8-month-old Holstein cattle raised on a rubber mattress were significantly higher than those of cattle raised on a concrete floor, but the difference in the ratio of feed to gain (F/G) was not significant [32]. The average daily gain of Charolaise and Limousin cattle raised on a rubber mattress was significantly higher than that on the concrete floor [12,33,34]. Our results were consistent with these previous reports. In this study, compared with that of the CF group, the average daily gain of the DFB group increased by 13.63%, and that in the SFB group increased by 7.21%, but the difference between them was not statistically significant. The ratio of feed to gain (F/G) of both the DFB group (20.93%) and the SFB group (16.76%) was significantly lower than that of the CF group. However, there was no significant difference in the F/G ratio between the SFB group and the DFB group. The results showed that soft fermented bedding played a positive role in the growth and development of bulls and it increased the feed conversion rate (F/G ratio), and the effect of deep fermented bedding was better than the shallow one. 

A previous study showed that cows prefer straw bedding to a rubber mattress [13]. Cows spend more time lying on deep sawdust and sand bedding (15 cm or above) than on shallow bedding (2–3 cm) [14]. Angus cattle spend significantly more time lying on a rubber mattress than on a concrete floor [19]. The total cow lying time, lying frequency, and bedding utilization rate by cows were significantly higher on deep sand bedding than on rubber mats. The preparation time before lying and standing time were shorter, indicating that cows were more comfortable lying on the deep sand bedding [23]. Besides, the addition of a large amount of sawdust bedding material onto geotextile mattresses positively affected cows’ lie-down preference [15]. Each additional kilogram of sawdust or straw to a stand resulted in an increase in cows’ lie-down time by 12 min per day [15]. Consistently, Drissler et al. [16] found that every 1 cm decrease in bedding thickness led to the reduction in cows’ lying time by 11 min every 24 h, and cows lay 1.15 h longer per day on thick bedding than on thin bedding. Compost-bedded pack had a positive impact on the welfare and comfort of cows [17,18]. 

Our results showed that the lying time and lying frequency of the DFB group and the SFB group were significantly longer and higher than those of the CF group, respectively. The preparation time before lying in the two fermented bedding groups was shorter than that in the CF group, and the lying time of the DFB group was significantly longer than that in the SFB group. Accordingly, the standing time of the two fermented bedding groups was significantly shorter than that of the CF group. The standing time of the DFB group was significantly shorter than that of the SFB group. This might be an essential reason for the higher gait score and joint injury degree in the CF group than in the DFB and SFB groups.

In this study, we observed differences in feeding and ruminant behaviors of bulls. At the same amount of feed intake, the daily feeding time and feeding frequency of bulls in fermented bedding groups were significantly shorter and less than those in the concrete floor group, respectively. However, there were no significant differences in feeding time and feeding frequency between the DFB and SFB groups. The rumination time and frequency of the two fermented bedding groups were significantly longer and higher than those in the concrete floor group, and those of the DFB group were significantly higher than those in the SFB group. These results indicated that the CF group had low feeding efficiency and spent more time standing for feeding and rumination. The fermented bedding groups spent less time consuming the same amount of feed and more time lying on bedding resting and ruminating. With the increase in rumination time, the amount of the secreted saliva would increase, and the saliva contains a buffer to alleviate rumen pH, thus improving digestion [35]. The reduction in rumination time would result in stress, anxiety, and disease to animals [36]. Furthermore, the energy consumption of cattle when they stand is higher than when they lie down [37], which explains why the average daily gain and feed utilization rate of the two fermented bedding groups are higher than the CF group in the case of no difference in feed intake between the fermented bedding groups and the concrete floor group. 

The body hygiene of bulls is one of the vital animal welfare indicators, and good body hygiene is important to the growth of bulls. Previous studies have shown that cows raised on deep sand bedding have better hygiene than ordinary straw bedding [31]. Using recycled manure solids as bedding material, the body hygiene of cattle in the deep bedding group was better than that in the shallow bedding group, but the difference was not significant [7]. In this study, the body hygiene of bulls in the two fermented bedding groups was significantly better than that in the concrete floor group, and the DFB group was significantly better than the SFB group. The reason might be that the fermented manure bedding had good absorption capacity to absorb the dung and urine of bulls. In addition, treading and plowing can mix fresh feces and urine, which is conducive to the decomposition of organic matter in the feces and urine by microorganisms from the bedding materials, and treading and plowing can also accelerate the evaporation of water in the bedding. 

Locomotion scores can be used as a reference to diagnose early lameness in cattle for the purpose of management adjustments [30]. Previous studies have shown that bulls have a lower occurrence of joint injury on rubber mats than on concrete floor [12,19]. The use of deep wood chips or straw bedding is preferred by cows and it can reduce hock joint lesions [20]. Consistently, cattle raised on deep sand bedding or deep recycled manure bedding exhibited fewer hock injuries and hock wear than those raised on rubber mattresses [8,22]. The detection of serum biomarkers of joint injury indicated that cattle hock joint injury on the concrete floor was significantly more severe than that on the deep sand bedding or deep fermented bedding [28]. Our experiment data revealed that the locomotion score of all the cattle in the SFB group and the DFB group was one until the end of the experiment with no serious hock joint injury observed in all cattle of these two groups. In the SFB group, two cattle had minor hock joint injuries. In the DFB group, only two cattle had slight hock joint injuries. However, in the CF group, one cow was severely lame (5 points), three cattle were moderately lame (3 points), two cattle were slightly lame (2 points), and eight cattle exhibited minor hock joint injuries. These results indicate that the deep fermented bedding could effectively reduce hoof injury and improve the welfare of bulls.

The main sources of carbon dioxide in barns are animal respiration, fecal discharge, and heating equipment at low temperatures [38]. The ammonia gas in barns has two major sources. One is ammonia generated by the gastrointestinal tracts of livestock, and the other is ammonia generated by microbial decomposition of organic matter such as feces, urine, and feed residue [39]. With the increase in ammonia and carbon dioxide concentrations in barns, bulls show restlessness, and they spend more time standing up and less time lying down [40]. With the increase in bedding thickness, bedding moisture content decreased, thus decreasing the ammonia content and carbon dioxide content in the bedding [41]. This is consistent with our results that at three time points (at 7:00, 12:00, and 18:00), the ammonia concentration and carbon dioxide concentration in the barn of the fermented bedding groups were significantly lower than those in the barn of the concrete floor group, and that with the increasing bedding thickness, the ammonia and carbon dioxide concentrations in the barn were decreased (Table 5).

## 5. Conclusions

Under low temperatures in winter, both deep fermented bedding and shallow fermented bedding exhibited a significant promoting effect on the growth of bulls. The DFB group had a significantly higher daily weight gain than the SFB group. Compared with shallow fermented bedding, deep fermented bedding increased feed utilization, thus reducing feed cost. Compared with the concrete floor, fermented manure bedding improved the welfare of bulls, and deep fermented bedding displayed a more obvious improvement effect on bull behavior and body hygiene than the shallow fermented bedding. Deep fermented bedding significantly reduced bull joint and hoof injuries. Meanwhile, both the DFB and SFB effectively improved the barn gas environment, and the improvement effect of the DFB was more obvious than the SFB. Our findings provide a reference for the utilization of bovine manure resources towards the harmonious development of the cattle breeding industry and the ecological environment.

## Figures and Tables

**Figure 1 animals-12-00925-f001:**
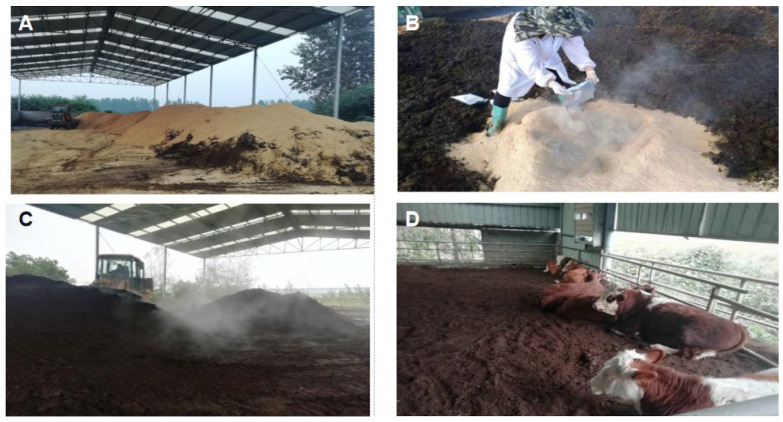
The production process of harmless fermentation bedding material of bovine manure. The manufacturing process of cow manure into bedding through a harmless fermentation treatment. (**A**) Collect fresh cow dung. (**B**) Addition of microbial species followed by even turning over of the pile. (**C**) Turning the pile regularly. (**D**) Bedding material distributed in the barn.

**Figure 2 animals-12-00925-f002:**
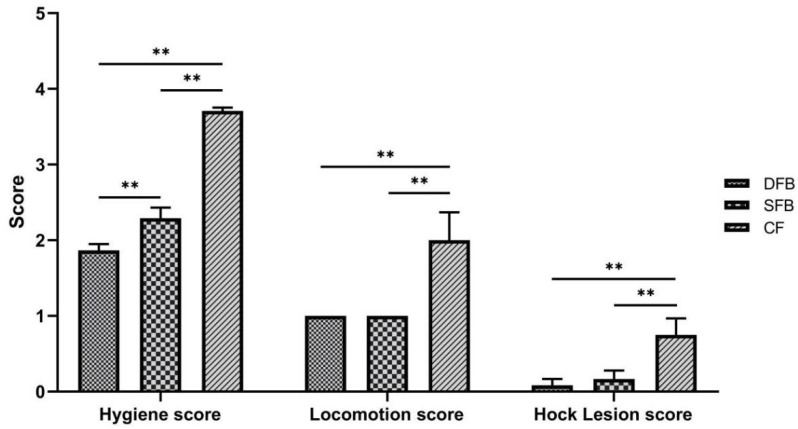
Effects of bedding thickness on animal welfare indices of bulls. The hygiene score was measured on the basis of a 4-point scale (1–4). The hock lesion score was measured on the basis of a 6-point scale (0–5), and the locomotion score was measured on the basis of a 5-point scale (1–5). Abbreviations: DFB, deep fermented bedding; SFB, shallow fermented bedding; CF, concrete floor. Values are presented as mean ± SEM and significant differences were displayed with either ** (*p* < 0.01).

**Table 1 animals-12-00925-t001:** Dietary composition and nutrient level (dry matter basis) during the feeding period.

Ingredients	Content (%)	Nutrients	Content (%)
Corn	25.00	Dry matter (DM)	52.01
Soybean meal	26.00	Crude protein	15.31
Vinasse	20.00	Crude fiber	5.27
Wheat bran	15.75	Crude fat	5.20
NaCl	3.30	Crude ash	5.19
Ca_3_(PO_4_)_2_	3.30	Calcium	0.84
NaHCO_3_	1.65	Phosphorus	0.70
Premix	5.00		
Total	100.00		

**Table 2 animals-12-00925-t002:** Basic information of detectors.

Name	Brand	Model	Detection Range (ppm)	Accuracy (ppm)
Portable carbon dioxide tester	Taiwan Hengxin	AZ77535	0–9999	30
Portable ammonia meter	China XiMa	AR8500	0–100	0.1

**Table 3 animals-12-00925-t003:** Effects of bedding thickness on growth performance of bulls.

Item	Bedding Treatment			SEM	*p*-Value
	DFB	SFB	CF		
Initial weight (Kg)	271.96	267.58	270.38	6.57	0.97
final weight (Kg)	343.83	334.58	332.50	7.26	0.80
ADG (Kg)	1.42	1.34	1.25	0.31	0.68
ADFI (Kg)	10.35	10.15	10.91	0.21	0.32
F/G	7.37 ^b^	7.76 ^b^	9.31 ^a^	0.33	<0.05

Abbreviations: DFB, deep fermented bedding (30 cm); SFB, shallow fermented bedding (15 cm); CF, concrete floor (0 cm); SEM, standard error of means; a,b mean in the same row with different superscripts represents a significant difference (*p* < 0.05).

**Table 4 animals-12-00925-t004:** Effects of bedding thickness on behavioral indices of bulls.

Item	Bedding Treatment			SEM	*p*-Value
	DFB	SFB	CF		
Lying duration (min/8 h)	223.14 ^a^	202.11 ^b^	125.42 ^c^	3.92	<0.01
Lying frequency (time/8 h)	4.23 ^a^	4.04 ^a^	1.69 ^b^	0.10	<0.01
Preparation time before lying (s)	23.66 ^b^	29.01 ^b^	41.74 ^a^	1.62	<0.01
Standing duration (min/8 h)	256.89 ^c^	280.27 ^b^	354.58 ^a^	3.98	<0.01
Standing frequency (time/8 h)	4.71 ^a^	4.50 ^a^	2.51 ^b^	0.10	<0.01
Intake duration (min/8 h)	101.35 ^b^	102.18 ^b^	124.20 ^a^	2.13	<0.01
Intake frequency (time/8 h)	4.75 ^b^	5.11 ^b^	7.56 ^a^	0.14	<0.01
Ruminate duration (min/8 h)	141.36 ^a^	115.94 ^b^	78.15 ^c^	2.48	<0.01
Ruminate frequency (time/8 h)	4.79 ^a^	4.04 ^b^	3.02 ^c^	0.09	<0.01

Abbreviations: DFB, deep fermented bedding (30 cm); SFB, shallow fermented bedding (15 cm); CF, concrete floor (0 cm); SEM, standard error of means; ^a–c^ mean in the same row with different superscripts represents a significant difference (*p* < 0.05).

**Table 5 animals-12-00925-t005:** Effects of bedding thickness on the barn environment of bulls.

Item	Time	Bedding Treatment			SEM	*p*-Value
		DFB	SFB	CF		
CO_2_ (ppm)	7:00	513.73 ^a^	515.64 ^a^	532.83 ^a^	6.21	0.38
	12:00	467.48 ^b^	480.64 ^b^	495.92 ^a^	4.01	<0.01
	18:00	506.54 ^a^	534.33 ^a^	558.23 ^a^	10.06	0.11
NH_3_ (ppm)	7:00	0.84 ^b^	0.96 ^b^	1.24 ^a^	0.06	<0.01
	12:00	0.85 ^b^	1.06 ^b^	1.44 ^a^	0.06	<0.01
	18:00	1.22 ^b^	1.55 ^b^	1.85 ^a^	0.13	<0.05
Temp (°C)	7:00	11.09 ^a^	10.03 ^b^	7.75 ^c^	0.22	<0.01
	12:00	10.92 ^a^	9.64 ^b^	8.42 ^c^	0.17	<0.01
	18:00	11.10 ^a^	9.97 ^b^	8.59 ^c^	0.17	<0.01

Abbreviations: DFB, deep fermented bedding (30 cm); SFB, shallow fermented bedding (15 cm); CF, concrete floor (0 cm); SEM, standard error of means; ^a–c^ mean in the same row with different superscripts represents a significant difference (*p* < 0.05).

## Data Availability

The datasets used and or analyzed during the current study are available from the corresponding author.

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
