# Peer review of "Effects of Fermented Manure Bedding Thickness on Bulls’ Growth, Behavior, and Welfare as Well as Barn Gases Concentration in the Barn"

_animals, 2022, doi:10.3390/ani12070925_

Round 1

Reviewer 1 Report

This paper could be interesting for readers of journal Animals. However it needs to be considerably improved. In many parts it is difficult to understand the meaning of sentences because of the poor English. I suggest first of all to rewrite the text in an appropriate English. then this paper can be evaluated again. During the revision of English, you need to take into consideration the comments I made in the pdf file (attached). 

Author Response

Dear Reviewer

Special thanks to you for your good comments.Those comments are all valuable and very helpful for revising and improving our paper, as well as the important guiding significance to our researches. I have responded to your comments on my manuscript (ID:Animals-1614789). My manuscript has been revised accordingly. Thank you for your valuable advice.

Reviewer 2 Report

Effects of Bedding Thickness on Some Evaluation Indices of Bull Housing in Free-Stall Barn Bedding with Fermented Manure

The topic is interesting and worth investigation. Despite the interesting topic, the authors however, should strictly follow the journal guideline for preparation of their manuscript. In order to comply with the formatting of the journal, I will point out some comments below. While the manuscript needs improvements in different sections, the originality and the novelty of this work make the readers interested.

Comments

Formatting issues:

Heading and subheading should be capitalized.

Table values should be consistent in decimal points (chose either one or two decimal points for values).

References should be formatted consistently and comply with the journal guideline

The template used is for 2021 but not 2022

“P” should be small and italic as I know. Please double check

Spaces are needed for “±” etc.

Lots of small grammatical and syntax errors. Lack of using “the” all over the text.

Section comments

Simple summary

L17: “were the best” is not scientific wording

L19: same as above comment

The last line is unclear

Abstract

Exceeding 200-word limit. Please double check

Conclusion line (last line) needs rewordings.

Introduction

L70: studies? Then cited only one study. Also, comma is not needed.

L73: The study also believed that….>> Another study showed that…

M&M

L118: please used past form of the verbs entirely.

The cowshed is semi-enclosed .. >>> The cowshed was semi-enclosed

L119: is>>>was

L122: is>>>was

L123: is>>>was

L123: is>>>was

L124: is>>>was

L125: ..two times >>> twice

Results

L179: Production Performance

Discussion

L288-295: Please use past verbs

>>>The results of this study showed that the lying time and lying frequency of bull in fermented bedding group were significantly higher than those in concrete floor group. The preparation time before lying was lower than that in CF group, and the lying time of bull in 290 DFB group was significantly higher than that in SFB.

L346-348: Studies? While cited one study?

L349>>> A research showed that…

Please find additional comments in the attachment.

Author Response

(The authors gave the same response as above.)

Reviewer 3 Report

This article compares deep and shallow depths of fermented manure tp concrete as a bedding option for bulls.

The paper was thorough in first explaining how manure is processed to be safe to be used for bedding. This was one of my initial questions/concerns for the welfare of animals lying in manure. 

Non-invasive measures were used to determine the welfare of the animals suggesting the deep fermented bedding improves welfare. The only addition I think that would be an improvement or a follow up to this paper is comparing the manure option to standard straw or sawdust on cattle welfare. Using a standard bedding option would be a good positive control treatment group to be able to say if manure improves welfare compared to that, and could discuss economics of using manure as an option if farms are able to compost what they already are collecting from their animals.  We know concrete can be detrimental to bone/joint health and is uncomfortable for cattle to stand/ly on, so it's not surprising the bedding did improve welfare. I'm at least glad to see this article compared different depths of bedding to indicate that that too makes a difference in welfare. 

Author Response

(The authors gave the same response as above.)

Round 2

Reviewer 1 Report

A minor revision, following all the comments present in the annexed text, has to be made

Author Response

Dear Reviewer

Special thanks to you for your good comments. Those comments are all valuable and very helpful for revising and improving our paper, as well as the important guiding significance to our researches. I have responded to your comments on my manuscript (ID:Animals-1614789). My manuscript has been revised accordingly. Thank you for your valuable advice.
